# Policy Optimized Text-to-Image Pipeline Design

**Uri Gadot[1,2]**      Rinon Gal [2]      Yftah Ziser [2,3]      Gal Chechik [2]      Shie Mannor [1,2]
[1]Technion      [2]NVIDIA Research      [3] University of Groningen

## Abstract

Text-to-image generation has evolved beyond single monolithic models to complex multi-component pipelines. These combine fine-tuned generators, adapters, upscaling blocks and even editing steps, leading to significant improvements in image quality. However, their effective design requires substantial expertise. Recent approaches have shown promise in automating this process through large language models (LLMs), but they suffer from two critical limitations: extensive computational requirements from generating images with hundreds of predefined pipelines, and poor generalization beyond memorized training examples. We introduce a novel reinforcement learning-based framework that addresses these inefficiencies. Our approach first trains an ensemble of reward models capable of predicting image quality scores directly from prompt-workflow combinations, eliminating the need for costly image generation during training. We then implement a two-phase training strategy: initial workflow vocabulary training followed by GRPO-based optimization that guides the model toward higher-performing regions of the workflow space. Additionally, we incorporate a classifier-free guidance based enhancement technique that extrapolates along the path between the initial and GRPO-tuned models, further improving output quality. We validate our approach through a set of comparisons, showing that it can successfully create new flows with greater diversity and lead to superior image quality compared to existing baselines.

## 1 Introduction

Recent advancements in generative AI have significantly improved the quality and diversity of text-to-image generation. Early models relied on monolithic architectures, where a single neural network directly translated textual prompts into visual outputs. However, as the field matured, it became clear that combining multiple specialized components—such as fine-tuned diffusion models, super-resolution modules, or specialized embeddings, into more sophisticated workflows leads to superior image quality and greater creative control [5, 40, 63]. This shift from monolithic models to modular workflows has been supported by user-friendly platforms such as ComfyUI[1], a popular open-source tool that allows users to visually construct complex generative pipelines through interconnected nodes represented in JSON format. ComfyUI has rapidly gained popularity due to its intuitive node-based interface, enabling users to assemble diverse generative models (e.g., Stable Diffusion, ControlNet, LoRAs) into flexible workflows tailored to specific image-generation tasks. Despite its accessibility, designing effective workflows remains challenging due to the vast space of possible component combinations and their prompt-dependent effectiveness. Consequently, crafting high-quality workflows typically requires considerable expertise and manual experimentation.

To address this challenge, recent work introduced ComfyGen [16], which uses large language models (LLMs) to automate the construction of prompt-adaptive workflows within ComfyUI. However, a key limitation of ComfyGen was its inability to generate genuinely novel workflow structures. At its core, their approach required synthesizing images using an extensive collection of pre-defined workflows

39th Conference on Neural Information Processing Systems (NeurIPS 2025).

---

[1]https://www.comfy.org/

and prompts, an expensive process limiting their training set's size. Constrained by this small set, their approach essentially learned a classifier over existing flows rather than synthesizing original graph topologies or selecting novel model combinations. This limitation significantly constrains the potential creativity and adaptability of automated workflow generation systems and, as we later show — may also limit their downstream performance.

In parallel, reinforcement learning (RL) has emerged as a powerful paradigm for fine-tuning large language models (LLMs), enabling them to optimize their outputs directly based on reward signals derived from human preferences or other evaluative metrics. Techniques such as Reinforcement Learning from Human Feedback (RLHF) have demonstrated remarkable success in aligning model behaviors with human expectations by iteratively refining model parameters based on explicit reward feedback. Furthermore, recent developments like Group Relative Policy Optimization (GRPO) introduced memory-efficient RL algorithms capable of optimizing policies without separate value functions, making them particularly suitable for complex sequential decision-making tasks. Building on these advancements, we propose FlowRL, a novel extension that integrates reinforcement learning into the workflow prediction framework to overcome its originality limitations. Specifically, we formulate workflow generation as an RL problem where an LLM-based policy sequentially constructs workflow graphs by selecting nodes and connections conditioned on textual prompts. To efficiently guide this process without incurring prohibitive computational costs associated with direct image generation for each candidate workflow during training, we introduce a surrogate reward model trained to predict image quality scores directly from prompts and workflow structures.

Finally, we adopt GRPO combined with per-token reward attribution mechanisms to provide granular feedback during policy updates. This affords our RL agent greater precision in identifying decisions within a generated workflow that contribute positively or negatively toward overall image quality.

In summary, our contributions are as follows

- We introduce ComfyGen-RL, the first RL-based approach for generating genuinely novel ComfyUI workflows tailored to align with human preference feedback.
- We propose a surrogate human-preference reward model enabling efficient RL training without computationally expensive image generations.
- We integrate GRPO with per-token reward attribution for stable and memory-efficient policy optimization.

Through these innovations, FlowRL significantly advances automated workflow generation capabilities, enabling richer creativity and greater adaptability in text-to-image synthesis pipelines.

## 2 Related Work

**Workflow Generation**

A recent line of research explores the use of compound systems, where multiple models or modules are chained together, often yielding superior performance compared to isolated models. These multi-component systems have been applied across fields ranging from programming challenges [1] and olympiad-level mathematics [53] to medical diagnostics [38] and video generation [64]. However, building compound systems presents significant challenges. Models must be chosen not only for their individual strengths, but also for their ability to complement each other. Moreover, the parameters of the different components should be selected with the entire system in mind. To address these difficulties, recent work has explored meta-optimization frameworks, where the structure and parameters of entire pipelines are automatically tuned for downstream performance [28]. Others have adopted graph-based architectures allowing dynamic reconfiguration of component interactions [68].

In the realm of text-to-image generation, recent work explores the use of pipelines using agentic systems [67, 61, 23], genetic algorithms [51] or by fine-tuning LLMs using large flow datasets tagged with human preference scores [16]. Although the human preference-based framework has shown promising results, it relies on creating and ranking images using large sets of flows. This, in turn, leads to challenges in effectively scaling the dataset and to a lack of ability to synthesize unseen flows at inference time. Our work aims to address this challenge by leveraging a policy-optimization approach for more effective exploration of the flow parameter space, coupled with a surrogate reward function which avoids the need to generate and rank a large set of images.

**Fine-Tuning LLMs with RL:** Reinforcement learning (RL) has become increasingly central to the development of large language models (LLMs), playing a key role in aligning model outputs with user preferences and enhancing task-specific capabilities. A prominent example is Reinforcement Learning from Human Feedback (RLHF) [39], which fine-tunes models using reward signals derived from human preferences to better align with communicative goals and social norms [8, 24]. Beyond alignment, RL has shown promise in improving LLMs' performance on domains requiring precise reasoning, such as mathematics [54, 56, 34] and code generation [30, 33]. Recently, [48] proposed Group Relative Policy Optimization (GRPO) as a scalable alternative to Proximal Policy Optimization (PPO). GRPO removes the need for a critic model by optimizing contrastive objective based on intra-group ranking, yielding better sample efficiency, improved stability, and reduced computational complexity [36, 46]. GRPO-trained LLMs demonstrated state-of-the-art performance in mathematical problem solving and code generation, highlighting its effectiveness on tasks requiring structured reasoning and adherence to correctness [48].

**Improving Text-to-Image Generation Quality**    The rapid adoption of text-to-image models [45, 37, 44, 13, 41] has led to many research efforts focused on improving their image quality and better matching human preferences. Some works focus on inference-time modifications, either optimizing noise seeds towards better behaving regions of the diffusion space [14, 43] or applying self-guidance and frequency-based modulations [21, 49, 35] to the generated features.

More commonly, models are tuned to provide better quality outputs. This is often done through carefully selected high-quality datasets or better captioning methods [9, 3, 47]. Another approach uses reward models [29, 59, 60, 31] to guide the generation process. These reward models can be used with reinforcement learning [4, 11, 15, 66], or through direct optimization [6, 42, 55].

Finally, recent methods explore the use of LLMs to improve text-to-image generation [62], commonly by using them to construct workflows featuring multiple models or chained editing tools [67, 51, 16]. Our work similarly uses LLMs to construct workflows, but better aligns them to human preferences through the use of reward models coupled with a reinforcement-learning feedback mechanism.

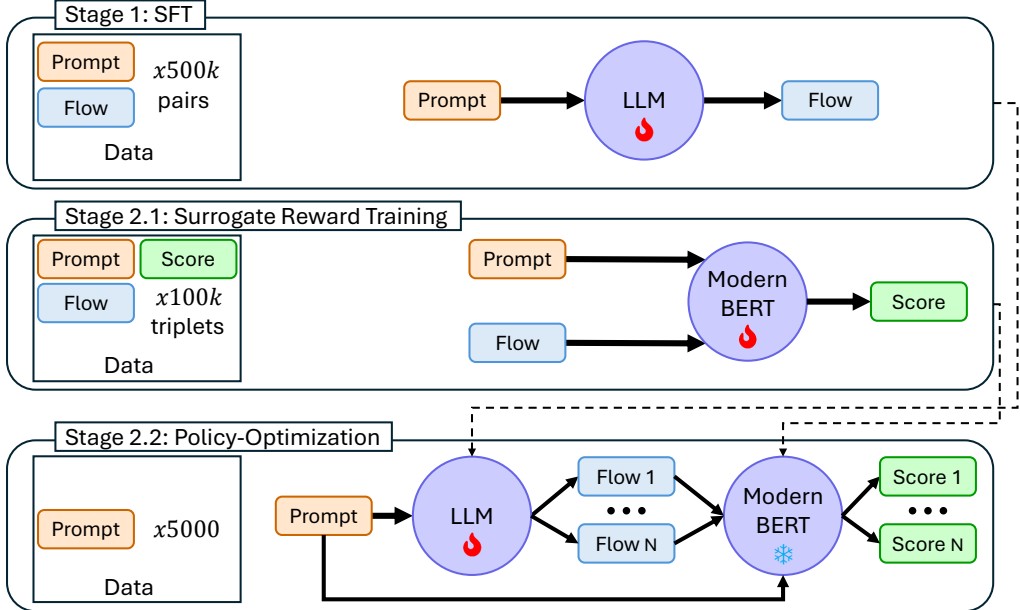

Figure 1: Pipeline overview. Step 1: Finetune LLM for general flow generation (SFT, 500K prompt-flow pairs). Step 2.1: Train reward model (100K prompt-flow-score triplets). Step 2.2: Optimize for quality using GRPO. 🔥 = learning, ❄ = frozen.

# 3 Methodology

Our goal is to enable efficient training of a human-preference based, prompt-to-workflow prediction system. Ideally, this system should be able to innovate and produce novel, unseen flows. Prior work struggled with this aspect, primarily due to their reliance on scoring images generated with a large set of fixed flows, whose parameters were sampled uniformly from a predefined set of options. To overcome this hurdle, we propose a two-phase training strategy. In the first, we pre-train on a large set of un-scored flows. This avoids the need to generate and score images, allowing us to use a much larger set to teach the LLM the structure of flows and the available components. Then, we perform a second tuning stage, where we leverage human-preference predictor models jointly with recent reinforcement-learning ideas (GRPO [48]) to drive the model towards better-performing subsets of the flow space. As training progresses, more samples are drawn from these regions, and hence, less computation is wasted on inefficient exploration.

However, generating and scoring images during LLM training is itself a costly process, which requires an order of a minute for every training step. Hence, we draw on ideas from the autonomous driving literature, where costly simulations are often replaced by faster predictors trained to replicate simulation outputs [2, 25, 26]. Here, we apply this idea by learning surrogate reward models that predict the final image score directly from the prompt and workflow pair. Notably, prior work has observed that such surrogates are susceptible to reward-hacking solutions [17, 50, 58]. Motivated by findings that ensembles can mitigate reward hacking [7, 65], we train an ensemble of such models and use their variance as a measure of uncertainty, allowing us to filter out samples that optimize for any individual surrogate reward model. Below we present these core components in greater detail and provide an overview of additional design choices or components that allow us to increase efficiency further or refine our results. An overview of our training pipeline is shown in Figure 1.

## 3.1 Training Data

To train our model we use the flow and prompt dataset of ComfyGen [16]. This set contains 33 human-created flows that define an overall graph structure, further augmented by randomly sampling novel parameter choices for existing blocks such as different base models, differnet LoRAs, diffusion samplers or even the number of steps and guidance scale. Since we do not need to score images for our first stage, we can apply more extensive augmentations and create $2,000$ variants from each baseline flow structure (compared with ComfyGen's 100). The set also contains 10000 prompts taken from the generation sharing website CivitAI.com. We keep the 500 prompts used to test ComfyGen as a holdout, and train using the rest.

## 3.2 Stage 1: Supervised Fine-Tuning on Flow Dataset

The first stage involves supervised fine-tuning (SFT) an LLM on a dataset of prompt-flow pairs without explicit score labels. At this stage, our goal is to teach the LLM the appropriate vocabulary and flow structure while maintaining output diversity. Our flow dataset $D_{SFT}$ consists of pairs $(p_i, f_i)$ where $p_i$ represents a randomly sampled prompt and $f_i$ represents a randomly sampled flow. We tune the model to take the sampled prompt $p_i$ and return its matching flow $f_i$. The full LLM query is shown in the supplementary. After fine-tuning, we evaluate the model's perplexity on $D_{SFT}$, achieving a score of $1.9$, which reflects strong alignment with the encoded workflows structural patterns.

**Efficient Flow Representation Scheme**  While prior work [16] directly predicts ComfyUI JSON representations, we note that these JSONs typically contain thousands of tokens, leading to long generation times and increasing memory requirements. An inspection of the tokenized JSONs shows that many tokens are wasted on maintaining the JSON format (e.g., on brackets or quotation marks) or on breaking down model or component names. Hence, to improve training efficiency and reduce token usage, we propose to modify the encoding scheme, using a novel structured representation that captures essential components while reducing token count. Additionally, we introduce specialized tokens to represent key elements of the flow. (e.g. tokens for ComfyUI node names or for model choices). An example of the difference between the two tokenization methods is outlined in Figure 2.

This new encoding scheme yields significant practical advantages resulting in substantial improvements in both computational efficiency and memory utilization. Quantitatively, the 86.7% reduction

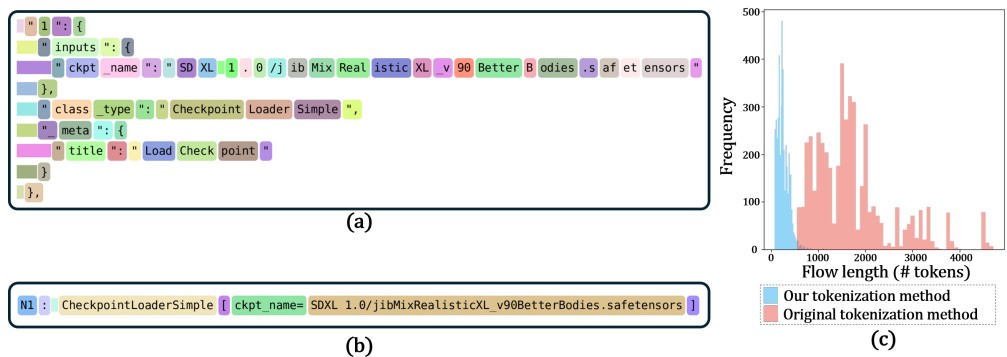

Figure 2: An example of a single ComfyUI node tokenized. **(a)** displays the original JSON input as tokenized by the standard Llama tokenizer. **(b)** shows our custom encoding, with introducing additional tokens to explicitly represent relevant components within workflow. Colored segment corresponds to a different token. **(c)** histogram of flows length (in token) of all training-set

in the average token length ($1500 \rightarrow 200$ tokens per workflow) enabled a $16\times$ batch size increase ($2 \rightarrow 32$ samples/batch) during the first-stage training. Ultimately reaching a $3\times$ time improvement over the original tokenization. These enhancements make it feasible to train complex models and apply memory-intensive algorithms, such as GRPO.

## 3.3 Stage 2: Reward-based policy-optimization

In the second stage, our goal is to tune the workflow-prediction LLM to better align it with flows that produce high quality outputs for a given prompt. To do so, we propose to leverage the recently introduced Group Relative Policy Optimization (GRPO) approach, which estimates advantages by comparing responses within groups of similar prompts, rather than relying on a separate value function. Using GRPO has two main benefits: (1) it eliminates the need to learn a separate value function, enabling better memory utilization during training and (2) its group-based reward normalization encourages greater exploration and diversity in generated workflows. However, the use of this approach requires us to score and rank the different candidate flows generated for each input prompt at training time. Naively, we could simply generate images with each such flow and score them using the human-preference predictors used by ComfyGen [16]. However, for complex flows, creating the images might take an order of a minute, greatly limiting the speed of training. Hence, we propose to avoid this lengthy generation step and instead train a surrogate reward model that will directly estimate the final reward from a pair of prompt and flow inputs.

**Surrogate Reward Model Training**  We implement the surrogate reward model $R_\phi$ on top of a ModernBert [57] backbone, with a novel output head trained to map the CLS token into a score. To tune the model, we feed it with strings containing a prompt and flow pair, and task it to predict the human-preference score for the image produced by this pair. For data, we use the ComfyGen dataset $D_R$, which contains triplets of prompt $p_i$, flow $f_i$ and score $s_i$. The surrogate's loss is then:

$$L_R(\phi) = \sum_{(p_i, f_i, s_i) \in D_R} MSE(R_\phi(p_i, f_i), s_i). \tag{1}$$

Although the construction of the original ComfyGen dataset still required generating images and scoring them, we find that the surrogate reward is much more sample efficient, performing well with just the 330 post-augmentation flows of ComfyGen (compared with our own $80k$ unscored flows).

### 3.3.1 Component-Aware Hybrid Reward Formulation

Since downstream flow performance can be heavily influenced by relatively few tokens (model choices, existence of specific blocks), we propose to further refine our surrogate model with a prefix-prediction score that is better able to assign credit to specific components. Specifically, we tune an additional reward model $R_\phi^{prefix}$ to predict the generated image score even when presented only with

randomly sampled prefixes of the flow:

$$L_{R^{pre}}(\phi) = \sum_{(p_i, f_i[1:j], s_i) \in D_R} MSE(R_\phi^{pre}(p_i, f_i[1:j]), s_i). \tag{2}$$

Our final reward design combines these two complementary signals to assign a different reward to each token $t$, depending on both the expected performance of the full flow, as well as a prefix ending with its component:

$$R(t) = R_\phi(p, f) + \sum_{j=1}^{J} \mathbb{1}_{t \in T_j} \cdot R_\phi^{pre}(p, f_{1:j}), \tag{3}$$

where $T$ are the tokens comprising the same flow component as $t$, and we sum over the contribution of the entire component.

### 3.3.2 Uncertainty-Aware Reinforcement Learning

Finally, prior work [17, 50, 58] observed that the use of surrogate reward models can lead to reward hacking. To avoid this pitfall, we train an ensemble of $N$ surrogate models $\{R_{\phi_1}, R_{\phi_2}, ..., R_{\phi_N}\}$, each using a different split of our training data. The ensemble provides us with both a more robust mean prediction, as well as with an uncertainty estimate:

$$\mu(p, f) = \frac{1}{N} \sum_{i=1}^{N} R_{\phi_i}(p, f); \quad \sigma(p, f) = \sqrt{\frac{1}{N} \sum_{i=1}^{N} (R_{\phi_i}(p, f) - \mu(p, f))^2}. \tag{4}$$

We can then define an uncertainty-aware reward function:

$$R(p, f) = \begin{cases} \mu(p, f) & \sigma(p, f) \leq \tau \\ 0 & \sigma(p, f) > 0 \end{cases}$$

where $\tau$ is a threshold parameter. This pessimistic approach assigns zero reward to prompt-flow pairs with high uncertainty, preventing the model from optimizing specific subsets of the reward ensemble, or from drifting to regions where the surrogate's predictions are unreliable.

### 3.4 Dual model guidance

As an additional step, we propose that results may be further improved through the use of a novel inference mechanism inspired by classifier-free guidance (CFG, [20]). Specifically, we draw on recent work on image generation [27] which demonstrate that diffusion models can be guided by extrapolating the predicted scores along the direction from an under-trained version of the model, and the fully trained one. We propose to apply a similar idea here, where we consider both our policy-optimized model ($\mathcal{M}_{GRPO}$, stage 2) and its "undertrained" SFT version ($\mathcal{M}_{SFT}$, stage 1). At inference time, generations are sampled by interpolating the logits of of both models:

$$\log p_{CFG}(f_j | f_{<j}, p) = \log p_{SFT}(f_j | f_{<j}, p) + \gamma \big( \log p_{GRPO}(f_j | f_{<j}, p) - \log p_{SFT}(f_j | f_{<j}, p) \big) \tag{5}$$

where $p_{CFG}$ represent the sampling distribution, $p_{SFT}$ is the next-token distribution of stage 1 model and $p_{GRPO}$ is the next-token distribution of stage 2 model. Finally, $\gamma \geq 0$ controls the guidance strength. Unless otherwise noted, we use $\gamma = 1.5$.

## 4 Experiments

### 4.1 Comparisons

We follow [16] and compare our approach to a set of baselines across two main metrics: (1) The GenEval [18] benchmark which measures prompt-adherence by using object detection and classification modules to evaluate correct object generation, placement, and attribute binding. (2) Human preference, using the CivitAI prompt-set of ComfyGen [16]. For the latter, we evaluate our approach using both an automated preference metric (HPS v2, [59]) as well as a user study.

We compare our approach against the following types of baselines: (1) Fixed, monolithic models including: SDXL, popular fine-tuned versions thereof, and SDXL-DPO, which was directly fine-tuned with human preference data. (2) Fixed, popular workflows, where we use the same workflow to generate all images regardless of the prompt. (3) Prior pipeline construction approaches, including agentic workflows that select and use off-the-shelf editing tools to correct generated content (GenArtist, [67]) and reward-based fine-tuned LLMs (ComfyGen [16]).

**Prompt adherence:** As summarized in Table 1, FlowRL demonstrates strong performance on the GenEval benchmark despite not being explicitly trained for prompt adherence. It achieves an overall score of 0.61, matching the best-performing baseline, ComfyGen. Notably, our approach outperforms other methods in the "two objects" (0.85 vs. 0.82) and "binding" (0.38 vs. 0.29) categories, indicating improved capability in handling complex compositional prompts. A representative qualitative example illustrating prompt adherence is provided in Figure 4.

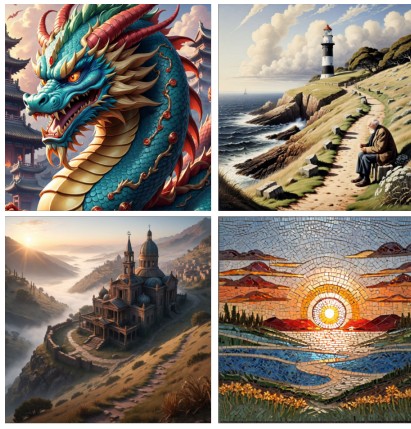

Figure 3: Example of generations with FlowRL

**Visual Quality:** To automatically evaluate the visual quality of FlowRL's outputs, we follow [55, 43, 16] and use a pair-wise comparison of HPS v2 [59] score between FlowRL and each baseline and report the average win rate. These comparisons use the full CivitAI test set of [16]. The win-rate of each baseline over FlowRL is reported in Table 1. Additionally, we conducted a user study were we show users 35 randomly sampled prompts and the images generated for each, using FlowRL and one of the baselines. Here, we focus on the best performing baseline from each category, as well as ComfyGen [16]. We then ask them to select the image that they prefer, taking both prompt adherence and visual quality into account. We report the aggregated win percentage in figure 5, and add more details in the supplementary. This experiment demonstrates FlowRL's capability to create more performant ComfyUI workflows for the given input prompts. Representative qualitative comparisons highlighting these improvements are provided in Figure 4, where our outputs consistently exhibit better prompt alignment and structural coherence compared to baseline generations.

| Model | Single object | Two object | Counting | Colors | Position | Attribute binding | Overall | HPSv2 winrate vs. FlowRL |
|---|---|---|---|---|---|---|---|---|
| SDXL | 0.98 | 0.74 | 0.39 | 0.85 | 0.15 | 0.23 | 0.55 | 2% ± 0.6% |
| JuggernautXL | **1.00** | 0.73 | 0.48 | 0.89 | 0.11 | 0.19 | 0.57 | 5%± 1% |
| DreamShaperXL | 0.99 | 0.78 | 0.45 | 0.81 | **0.17** | 0.24 | 0.57 | 3%± 0.6% |
| DPO-SDXL | **1.00** | 0.81 | 0.44 | **0.90** | 0.15 | 0.23 | 0.59 | 5%± 1% |
| Most Popular Flow | 0.95 | 0.38 | 0.26 | 0.77 | 0.06 | 0.12 | 0.42 | 13%± 1% |
| 2nd Most Popular Flow | **1.00** | 0.65 | **0.56** | 0.86 | 0.13 | 0.34 | 0.59 | 14%± 1% |
| GenArtist | 0.94 | 0.41 | 0.40 | 0.72 | **0.24** | 0.07 | 0.47 | 5% ± 1% |
| RPG-DiffusionMaster | **1.00** | 0.64 | 0.21 | 0.89 | 0.20 | 0.35 | 0.55 | 3%± 0.8% |
| ComfyGen | 0.99 | 0.82 | 0.50 | **0.90** | 0.13 | 0.29 | **0.61** | 40% ± 2% |
| FlowRL (Ours) | **1.00** | **0.85** | 0.44 | 0.86 | 0.11 | **0.38** | **0.61** | - |

Table 1: GenEval and HPS v2 comparisons. FlowRL is on-par with ComfyGen on GenEval and outperforms all other baseline approaches in overall score. On human preference metrics, FlowRL significantly outperforms prior methods. CIs are calculated as one standard deviation from the mean.

**Novelty of generated flows:** A key advantage of our approach lies in its capacity to generate workflows that are not merely copies of those seen during training. To quantify this novelty, we generate 500 flows using the CivitAI test set, and calculate the normalized Levenshtein distance (NLD) [52, 32] between each generated workflow and its nearest training sample. We further normalize these values by the NLD between training samples, giving us a measure of what fraction of the variance in training data we manage to preserve. Additionally, we report how many generated flows exist "as-is" in the training data, and how many unique flows were created in the 500 output set.

The results are reported in Table 2. Our experiments confirm the findings of [16] which report that their approach learned to copy flows from the training data. FlowRL meanwhile achieves significantly

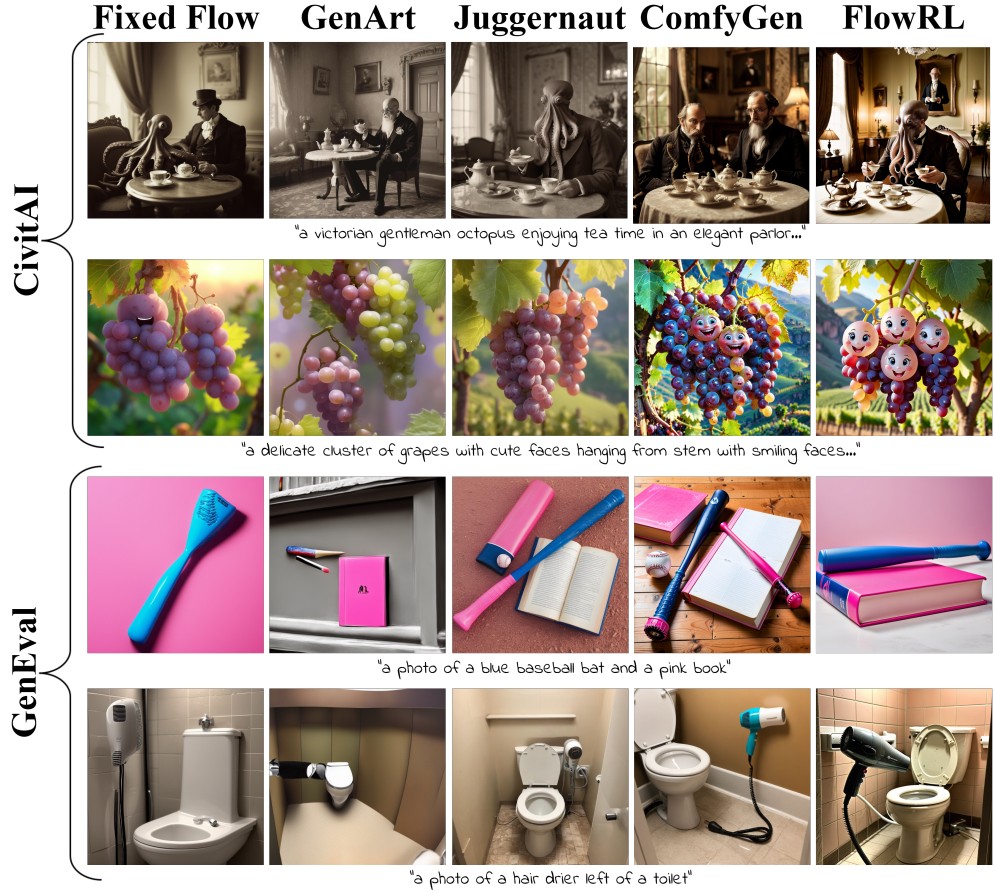

Figure 4: Qualitative results on CivitAI and GenEval prompts.

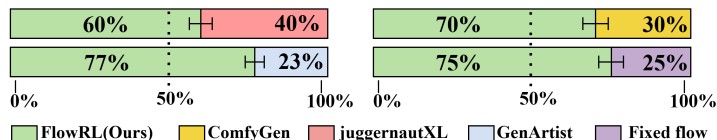

Figure 5: Human study win rate of FlowRL vs other relevant baselines

higher novelty, demonstrating the ability to generalize to new parameter combinations. These results highlight the effectiveness of our reinforcement learning framework in encouraging the LLM to explore and produce a broader range of complex workflows.

**Effects of Dual model guidance:** Next, we investigate the impact of our dual-model guidance approach. Specifically, prior work [12] highlighted the ability of guidance-based methods to trade diversity for performance (or recall for precision). We show that similar behavior can be observed here. As shown in Table 2, while increasing guidance strength ($\gamma$) improves HPS v2 scores win-rate vs ComfyGen, it significantly impacts the structural diversity of the generated workflows. At $\gamma = 1.5$, our method maintains the uniqueness of generated flows. However, as $\gamma$ increases, we observe a dramatic reduction in the uniqueness ratio to just 8%.

Notably, all our FlowRL variants maintain near-zero overlap with training data (0-1% "exists in data" vs ComfyGen's 94%), and the NLD ratio actually improves with guidance (from 0.6 without CFG to 0.75 at $\gamma = 2$). This pattern suggests that stronger guidance pushes the model to consistently generate a smaller subset of high-performing workflows, effectively concentrating probability mass on patterns that maximize reward but reducing exploration of the solution space. Conceptually, this mirrors

observations in image generation with CFG, where higher guidance strengths produce higher-quality but less diverse output.

| Method | unique ratio (%) | exists in data (%) | NLD ratio | HPSv2 win-rate Vs ComfyGen |
|---|---|---|---|---|
| ComfyGEN | 7% | 94% | 0 | - |
| FlowRL (w/o CFG) | 41% | 1% | 0.6 | 59% |
| FlowRL + CFG($\gamma = 1.5$) | 41% | 0% | 0.74 | 60% |
| FlowRL + CFG ($\gamma = 2$) | 8% | 0% | 0.75 | 63% |

Table 2: Comparison of originality of flow generation models

## 4.2 Ablation study

To quantify the impact of individual components in FlowRL, we conducted an ablation study comparing variants with and without our key improvements. We evaluated the following modifications: (1) removing the component-aware reward model, (2) removing the uncertainty ensemble cutoff, (3) varying number of BERT models in our reward ensemble, (4) dropping the SFT step (stage 1), and (5) dropping the GRPO-tuning step. For (5), we instead use the stage-1 model to sample five flows per prompt, and use our reward ensemble to score them in relation to the prompt. Then, we generate an image with the highest scoring flow. Finally, to ensure that our benefits are not grounded in the novel encoding scheme, we also evaluate a baseline ComfyGen [16] model trained on this new representation. We compare all scenarios against both the original ComfyGEN and against our full model, using HPSv2 scores on the CivitAI prompt set. The errors reported are the $1 - \sigma$ Wald interval.

| win ratio | w/o prefix reward | w/o reward cutoff | Ensemble of 1 | Ensemble of 3 Berts | Ensemble of 5 | ComfyGen (+ encoded) | SFT only | w/o SFT stage |
|---|---|---|---|---|---|---|---|---|
| vs ComfyGen(%) | 55 ±2.22 | 57 ±2.22 | 55 ±2.22 | 56 ±2.21 | 56 ±2.22 | 37 ±2.16 | 29 ±2.02 | 0 - |
| vs ours (%) | 42 ±2.21 | 45 ±2.21 | 33 ±2.1 | 34 ±2.12 | 36 ±2.15 | 26 ±1.96 | 19 ±1.75 | 0 - |

Table 3: The win ratio on the HPSv2 score for each component of our method compared to (1) the ComfyGen baseline and (2) the full ComfyGenRL model, using head-to-head comparisons.

The results are presented in table 3. These demonstrate the vital contribution of each component to overall performance. The full model consistently outperforms all ablations, with particularly significant drops observed when removing the SFT stage entirely (0% win rate against ComfyGen and our full model). This emphasizes the critical nature of proper initialization before applying reinforcement learning methods. Looking at specific components, "prefix reward" proves the most beneficial, showing the importance of assigning more granular rewards. The "ComfyGen (+encoded)" variant, which uses our encoding scheme but lacks reinforcement learning, achieves only a 37% win rate against the original ComfyGen, highlighting that our encoding improvements work synergistically with the GRPO training approach.

## 5 Discussion

This paper presents a novel approach for fine-tuning LLMs using a combination of supervised learning on flow data, surrogate reward modeling, and uncertainty-aware reinforcement learning. Our method addresses several key challenges in LLM fine-tuning, including reward hacking, distribution shifts, and training efficiency. The results demonstrate that our approach outperforms existing baselines across multiple metrics. Importantly, compared to prior workflow generation work, our approach demonstrates greater output diversity and successfully generalizes to novel flows that did not exist in the training data.

Although it improves on the current state-of-the-art in multiple aspects, our approach still maintains many of their limitations. First, it remains focused on text-to-image workflows, with no support for editing tasks or video modules. Second, introducing new workflow components to the LLM would require retraining our entire stack. In the future, we hope to explore more efficient ways of adapting to novel models or blocks.

By enabling reliable and diverse automated workflow generation, our work advances generative AI systems that adapt to human preferences. We hope it will help foster more collaborative innovation by streamlining the integration of independently trained, specialized modules.

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
