# OpenReview forum: "Policy Optimized Text-to-Image Pipeline Design"
_NeurIPS.cc/2025/Conference — NeurIPS 2025 poster_

### Official Review · Reviewer_9pZC · 2025-06-26

**Clarity:** 2
**Significance:** 2
**Originality:** 2
**Rating:** 2
**Confidence:** 3

**Summary:**

The paper introduces FlowRL, a reinforcement learning based framework for generating text-to-image workflows in ComfyUI. It uses RL-based approach for generating novel ComfyUI workflows aligned with human preferences and predicts image quality scores directly from prompt-workflow pairs, avoiding costly image generation during training. It also introduces dual-model guidance between SFT and GRPO-tuned models during inference.

**Questions:**

1. See weakness

2. I think the writing quality needs improvement. The article actually uses a lot of content to describe methods for solving engineering problems, but I believe its contribution in terms of methodological innovation is limited.

3. Some parameters in the article may require more detailed descriptions to make the paper easier to understand.

**Ethical Concerns:**

["NO or VERY MINOR ethics concerns only"]

**Final Justification:**

I appreciate the author’s efforts in addressing my concerns, but regarding the reinforcement learning optimization on workflows, I still have doubts about the contribution in terms of the algorithm.

**Limitations:**

Yes.

**Paper Formatting Concerns:**

No.

**Quality:**

2

**Strengths And Weaknesses:**

### Strengths

1. No need to generate images for scoring and obtaining rewards.
2. Tested across multiple metrics.

### Weaknesses
1. On line 156 of the paper, perplexity reaches 1.9, but the significance of this value is not explained—e.g., no comparison with other methods.
2. The third contribution of the paper: "We integrate GRPO with per-token reward attribution for stable and memory-efficient policy optimization." seems relatively weak. The paper achieves higher batch sizes and faster training speeds by compressing JSON format, which appears to be an optimization of dataset processing rather than a methodological contribution.
3. The paper mentions "flow," but could provide more examples—such as its format and components—for clarity.
4. One issue is that BERT learns a reward model for prompt-flow pairs, but the scores are assigned to generated images. A key question is whether higher scores for prompt-flow pairs (without image generation) equate to improved image quality. This point is not well demonstrated in the paper. Particularly in the Geneval evaluation, FlowRL performs worse than ComfyGen in counting, colors, and position.
5. The terms p_SFT and p_GPRO in Equation (5) are not defined in the paper.
6. The methodology primarily applies GRPO for reinforcement learning training of LLMs, introducing some training tricks (e.g., optimizing dataset format, per-token reward). However, these are largely engineering solutions for GRPO training. The main contribution—applying GRPO to T2I workflows—seems somewhat limited. To elevate the paper’s quality, more innovative elements should be incorporated.

---

> ### Author Rebuttal · Authors · 2025-07-31
>
> We thank the reviewer for the feedback on our paper. We address the reviewer concerns:
>
> **Q:**
> > "On line 156 of the paper, perplexity reaches 1.9, but the significance of this value is not explained—e.g., no comparison with other methods."
>
> **A:** The perplexity of 1.9 on line 156 indicates that our model is highly confident and accurate in predicting the next token in the workflow sequence, reflecting a strong understanding of the workflow structure and syntax. Which is an indication that the SFT stage is finalized and the optimization can be switched to the GRPO stage. We explored the option of optimizing the RL agent without the initial SFT stage in our ablation(section 4.2). We found this stage to be crucial due to the highly structured nature of the generated patterns, which necessitates its use as a prior.
>
> **Q:**
> > "The third contribution of the paper: "We integrate GRPO with per-token reward attribution for stable and memory-efficient policy optimization." seems relatively weak. The paper achieves higher batch sizes and faster training speeds by compressing JSON format, which appears to be an optimization of dataset processing rather than a methodological contribution."
>
> **A:** We respectfully clarify that while compressing the JSON format to our "encoded" format is an engineering optimization for efficiency, the "integration of GRPO with per-token reward attribution for stable and memory-efficient policy optimization" is indeed a methodological contribution. This approach was crucial for effectively applying GRPO to a sequential generation task like workflow design, where rewards need to be attributed to specific actions (tokens) within a long sequence. It addresses the challenges of credit assignment in complex, structured output generation, which goes beyond simple dataset processing.
>
> **Q:**
> >"The paper mentions "flow," but could provide more examples—such as its format and components—for clarity."
>
> **A:** We acknowledge that providing more detailed examples of flow formats and components could enhance clarity. We will add examples of workflows in the supplementary material of the final version.
>
> **Q:**
> >"One issue is that BERT learns a reward model for prompt-flow pairs, but the scores are assigned to generated images. A key question is whether higher scores for prompt-flow pairs (without image generation) equate to improved image quality. This point is not well demonstrated in the paper. Particularly in the Geneval evaluation, FlowRL performs worse than ComfyGen in counting, colors, and position."
>
> A: The surrogate reward model (BERT-based) is trained to predict generated image quality scores directly from prompt-workflow pairs without generating an image. Specifically, these scores are derived by generating images and scoring them with off-the-shelf human preference predictors (e.g., Pickscore[1], HPS[2], ImageReward[3]) which have been shown (in their own published papers) to correlate strongly with human preference. Our experiments also use such models to validate the quality of our final outputs, and this is further verified with a user study. These experiments demonstrate that optimizing with the learned reward model indeed leads to improved image quality. The GenEval metric, which focuses on prompt-following, does not measure image quality and may not fully capture these qualitative improvements.
>
> **Q:**
> >"The terms p_SFT and p_GPRO in Equation (5) are not defined in the paper."
>
> **A:** We apologize for this oversight. In Equation (5), $p_{SFT}$ refers to the policy learned during the Supervised Fine-Tuning (SFT) stage, and $p_{GRPO}$ refers to the policy optimized using Group Relative Policy Optimization (GRPO). We will ensure these terms are clearly defined in the main text of the final version.
>
> **Q:**
> >"The methodology primarily applies GRPO for reinforcement learning training of LLMs, introducing some training tricks (e.g., optimizing dataset format, per-token reward). However, these are largely engineering solutions for GRPO training. The main contribution—applying GRPO to T2I workflows—seems somewhat limited. To elevate the paper’s quality, more innovative elements should be incorporated."
>
> **A:** While GRPO is an existing algorithm, its application to the sequential generation of structured ComfyUI workflows is non-trivial. As our experiments demonstrated, our approach relies on components such as improved reward attribution and a surrogate reward ensamble design, which together represent a methodological advancement for a new, challenging problem space. Moreover, the use of GRPO allows us to bypass fundamental limitations (diversity) of the best performing methods in this problem space, representing not only a quantitative improvement, but a push for a better training approach.
>
> Finally, we note that studies on non-trivial adaptations of RL methods for image generation tasks are regularly published in top tier conferences (e.g., Diffusion Model Alignment Using Direct Preference Optimization, CVPR 2024[4]).
>
> **Q:**
> > "I think the writing quality needs improvement. The article actually uses a lot of content to describe methods for solving engineering problems, but I believe its contribution in terms of methodological innovation is limited." & "Some parameters in the article may require more detailed descriptions to make the paper easier to understand."
>
> **A:** We appreciate the reviewer feedback on writing quality and will review the paper for areas of improvement to enhance clarity and emphasize our methodological contributions more explicitly. We agree that more detailed descriptions of certain parameters could benefit reader understanding. We will ensure that additional details on key parameters are provided in the supplementary material of the final version to improve clarity.
>
> **References:**
>
> [1] Yuval Kirstain, Adam Polyak, Uriel Singer, Shahbuland Matiana, Joe Penna, and Omer Levy.
> Pick-a-pic: An open dataset of user preferences for text-to-image generation. In Thirty-seventh
> Conference on Neural Information Processing Systems, 2023.
>
> [2]Xiaoshi Wu, Yiming Hao, Keqiang Sun, Yixiong Chen, Feng Zhu, Rui Zhao, and Hongsheng
> Li. Human preference score v2: A solid benchmark for evaluating human preferences of
> text-to-image synthesis. arXiv preprint arXiv:2306.09341, 2023.
>
> [3]Jiazheng Xu, Xiao Liu, Yuchen Wu, Yuxuan Tong, Qinkai Li, Ming Ding, Jie Tang, and Yuxiao
> Dong. Imagereward: Learning and evaluating human preferences for text-to-image generation.Advances in Neural Information Processing Systems, 36, 2024.
>
> [4]Wallace, B., Dang, M., Rafailov, R., Zhou, L., Lou, A., Purushwalkam, S., Ermon, S., Xiong, C., Joty, S., & Naik, N. (2024). Diffusion Model Alignment Using Direct Preference Optimization. In Proceedings of the IEEE/CVF Conference on Computer Vision and Pattern Recognition (CVPR) (pp. 8228-8238).

---

> > ### Comment · Reviewer_9pZC · 2025-08-05
> >
> > Thank you for your response.
> >
> > Regarding RL methods for image generation tasks, I believe the direction of this article is different from that of Diffusion Model Alignment Using Direct Preference Optimization (Diffusion DPO). This article focuses on optimizing workflows, while Diffusion DPO uses DPO to train diffusion models. Although both involve reinforcement learning training, the training objectives are not the same.
> >
> > I appreciate the author’s efforts in addressing my concerns, but regarding the reinforcement learning optimization on workflows, I still have doubts about the contribution in terms of the algorithm.

---

> ### Author Response · Authors · 2025-08-06
> **Response to reviewer 9pZC comment**
>
> Thank you for your response and continued engagement with our work.
>
> Our reference to Diffusion DPO was not to claim an identical objective, but to demonstrate that focusing on applications by applying existing RL algorithms to novel problem spaces within the image generation domain is a valid and recognized contribution. Examples include, but are not limited to: using policy optimization for fine-tuning diffusion models to align with human preferences [1, 2], using an RL agent to control external image editing software [3], optimizing prompts for T2I models using RL [4] and more. These works appeared in top ML conferences such as CVPR and NeurIPS.
>
> In these cases, the core contribution lies in the non-trivial adaptation of an existing RL algorithm to solve a new problem with unique challenges. Our work makes a similar contribution by successfully apply an RL framework to the highly structured and sequential task of ComfyUI workflow generation.
>
> ___________
>
>
> References:
>
> [1] Wallace, B., et al. "Diffusion Model Alignment Using Direct Preference Optimization." Proceedings of the IEEE/CVF Conference on Computer Vision and Pattern Recognition (CVPR), 2024.
>
> [2] Xu, J., Liu, X., Wu, Y., Tong, Y., Li, Q., Ding, M., Tang, J., & Dong, Y. (2023). DPOK: Reinforcement Learning for Fine-tuning Text-to-Image Diffusion Models. In Proceedings of the 37th Conference on Neural Information Processing Systems (NeurIPS 2023). NeurIPS.
>
> [3] Kosugi, S., & Yamasaki, T. (2020). Unpaired Image Enhancement Featuring Reinforcement-Learning-Controlled Image Editing Software. In Proceedings of the AAAI Conference on Artificial Intelligence (AAAI). AAAI Press.
>
> [4] Hao, Yaru, et al. "Optimizing prompts for text-to-image generation." Advances in Neural Information Processing Systems 36 (2023): 66923-66939.

---

### Official Review · Reviewer_p5tr · 2025-07-01

**Clarity:** 3
**Significance:** 2
**Originality:** 2
**Rating:** 4
**Confidence:** 4

**Summary:**

This paper presents a policy optimization strategy for workflow prediction for text-to-image tasks. The idea drawing from previous work is to prompt/finetune an LLM to directly predict the ComfyUI workflow for the task of text-to-image generation. Similar to several LLM fine-tuning pipelines these days, this involves a supervised fine-tuning stage on a curated dataset of workflows, followed by training a surrogate reward model, and then a policy optimization stage (using GRPO) to maximize these rewards. Results on prompt following and other reward model evaluations indicates that the proposed method outperforms prior works exploring similar concepts for the task.

**Questions:**

There are 2 major aspects where I am curious to see if there are improvements possible:

a) Choice of Evaluation: While GenEval provides a clean benchmark for prompt following on relatively short and basic prompts, the ideal usecases of FlowRL/ComfyGen etc. are probably quite different. Would it be possible to have more meaningful evaluations of the proposed methods beyond the standard evaluations provided in the paper to the kind of tasks that an end user of such a framework might actually care about?

b) Novelty of Generated Flows/Source of Improvements: An interesting aspect of end-to-end optimization is that we might observe novel sources of improvements. While the analysis in Tab. 2 is a first step in this direction (observing the novel generated flows), it might be worthwhile to have a deeper dive into this to see the exact nature of these novel flows to see what causes the underlying improvements?

**Ethical Concerns:**

["NO or VERY MINOR ethics concerns only"]

**Final Justification:**

I would like to thank the authors for providing a clear clarification in the rebuttal. Having looked over the paper, the reviews, as well as the rebuttal, I would like to stick with my original rating (Borderline Accept). I do think that the paper approaches an important problem and provides valuable insights, and therefore merits acceptance to me, despite the fundamental method being a relatively direct application of GRPO for agentic text-to-image generation.

**Limitations:**

Yes.

**Quality:**

3

**Strengths And Weaknesses:**

Strengths:

The paper makes the natural and principled step towards end-to-end optimization of the workflow generation concept (as opposed to just prompting/in-context learning) with a reward ensemble which provides a more robust optimization strategy.

Further, the paper also has a robust ablation analysis covering important components of the proposed pipeline providing convincing results.

The paper also does a good job of presenting the paper in a clear and easy to understand manner.

Weaknesses:

The only major weakness I can see in the paper is that it is a "minimal" improvement over ComyGen in many aspects (the overall formulation, training data for SFT etc.). In terms of the overall improvements, while ComfyGen and the proposed FlowRL have the same GenEval results, the HPSv2 and the human study seem to indicate improvements for FlowRL over ComyGen. However, it might be a good idea to clarify the aspects in which FlowRL improves over ComfyGen and how these might not be captured by GenEval.


Overall, I do think that the paper broadly is a step in the right direction and therefore would merit acceptance to me, however I do think it would be a good idea to clarify both the comparisons, evaluations, and have better analysis on the source of improvements/novelty of generated workflows.

---

> ### Author Rebuttal · Authors · 2025-07-31
>
> We thank the reviewer for the comprehensive review and valuable feedback on our paper. We are encouraged by the recognition of our principled step towards end-to-end optimization of workflow generation for T2I tasks, and the robust ablation analysis which underscores the effectiveness of our approach. We address the the reviewer concerns below:
>
> **Q:**
> >  "Choice of Evaluation: While GenEval provides a clean benchmark for prompt following on relatively short and basic prompts, the ideal usecases of FlowRL/ComfyGen etc. are probably quite different. Would it be possible to have more meaningful evaluations of the proposed methods beyond the standard evaluations provided in the paper to the kind of tasks that an end user of such a framework might actually care about?" & "The only major weakness I can see in the paper is that it is a "minimal" improvement over ComyGen in many aspects (the overall formulation, training data for SFT etc.). In terms of the overall improvements, while ComfyGen and the proposed FlowRL have the same GenEval results, the HPSv2 and the human study seem to indicate improvements for FlowRL over ComyGen. However, it might be a good idea to clarify the aspects in which FlowRL improves over ComfyGen and how these might not be captured by GenEval."
>
>
> **A:** We acknowledge your observation regarding the comparison with ComfyGen. While GenEval results might appear similar, FlowRL's improvements are primarily evident in HPSv2 and the user study (70% win-rate in the user study, and 60% win-rate in the HPSv2 score). Those metrics better reflect overall image quality and aesthetic preference. GenEval, being a prompt-following metric, may not fully capture these qualitative improvements in image generation. In addition, while GenEval serves as a valuable academic benchmark with its simplified prompts, the CivitAI prompts are taken from images created and shared by real users, and thus offer a more representative evaluation of the model's performance in real-world user scenarios.
> The core improvement of FlowRL lies in its end-to-end policy optimization with a reward model, which allows it to discover more optimal and nuanced workflow configurations than ComfyGen's supervised learning approach. The "novelty of generated workflows" (as discussed in your second question) is a direct outcome of this optimization, leading to better-performing flows not present in the initial SFT dataset.
>
> **Q:**
> >  "Novelty of Generated Flows/Source of Improvements: An interesting aspect of end-to-end optimization is that we might observe novel sources of improvements. While the analysis in Tab. 2 is a first step in this direction (observing the novel generated flows), it might be worthwhile to have a deeper dive into this to see the exact nature of these novel flows to see what causes the underlying improvements?"
>
> **A:** We concur that a deeper dive into the exact nature of novel flows and their contribution to improvements is valuable. The GRPO stage inherently encourages exploration beyond the initial SFT dataset, allowing the policy to discover workflow configurations that yield higher rewards but were not explicitly present in the training data.
> To further explore the nature of these novel flows, we followed ComfyGen and conducted a TF-IDF analysis focusing on the specialized usage of different models across different categories of prompts. For example, we observed that *JibMixRealisticXL* dominates in Photo-realistic and People categories, *TurboVisionXL* was mostly used in darker, more technical themes such as Gothic and Horror, *CrystalClearXL* is stronger in artistic, creative, and abstract categories, *DreamShaperXL* is mostly used in natural and cosmin themes. In addition, *AlbedoBaseXL* is used in very specific and niche categories (such as Steampunk and Underwater). This analysis reveals distinct specializations and category clusters among the models used in the optimized flows, indicating how the policy leverages specific models for particular content generation needs. We are happy to add a supplementary table outlining the most prominent model choices for each prompt category.

---

### Official Review · Reviewer_2NbJ · 2025-07-03

**Clarity:** 3
**Significance:** 3
**Originality:** 3
**Rating:** 5
**Confidence:** 2

**Summary:**

The paper introduces FlowRL, a reinforcement learning based framework designed to automate the text-to-image generation workflow within the ComfyUI ecosystem. FlowRL is consisted of 2 stages: a supervised fine-tuning on dataset of unscored prompt-flow pairs to learn structural priors, and a RL optimization using group relative policy optimization (GRPO), which predict image quality directly from prompt-flow pairs instead of generated images. In addition, the model employs a classifier-free guidance mechanism at inference to balance quality and diversity. Experimental evaluations show that FlowRL outperforms baseline methods.

**Questions:**

- Can the authors elaborate on how transferable FlowRL is to other domains or platforms?
- How does the performance of FlowRL scale as the diversity of prompts and flow complexity increases? Is there evidence of reward model generalization in unseen domains or under distribution shifts?

**Ethical Concerns:**

["NO or VERY MINOR ethics concerns only"]

**Final Justification:**

The authors addressed my questions and I would love to keep my score.

**Quality:**

4

**Strengths And Weaknesses:**

Pros:
- The paper has strong empirical validation. The proposed method is comprehensively benchmarked against baselines with GenEval and HPSv2 metrics. The experiments show better performance than previous methods.
- The paper shows good technical innovation. The proposed RL framework overcomes the limitation of prior model (ComfyGen) and improves efficiency and scalability. By introducing a surrogate reward model ensemble, the authors effectively eliminate the need for image generation during training.
- The paper has good concept of framing of workflow generation as a sequential RL task using LLMs is a fresh and significant departure from prior classifier-based or static methods.

Cons:
- The method could have limited domain generalization since it is focused on ComfyUI-based text-to-image pipelines. The generalization to other domains is not tested and discussed in depth. Additionally, since the model builds heavily on the ComfyGen dataset, the reproducibility is limited.

---

> ### Author Rebuttal · Authors · 2025-07-31
>
> We thank the reviewer for the valuable feedback. We're pleased that the reviewer found our empirical validation strong and the technical innovation of our proposed RL framework to be novel. We address the concerns below:
>
>
> **Q:**
> >  "Can the authors elaborate on how transferable FlowRL is to other domains or platforms?" & "The method could have limited domain generalization since it is focused on ComfyUI-based text-to-image pipelines. The generalization to other domains is not tested and discussed in depth. Additionally, since the model builds heavily on the ComfyGen dataset, the reproducibility is limited."
>
> **A:** The trained FlowRL model is not directly transferable to other domains or platforms because it has learned the specific "language" (API, tokens) and structure of ComfyUI workflows, and was trained exclusively on ComfyUI flow data. However, the core framework – using reinforcement learning for pipeline design with a surrogate reward model – is conceptually transferable to other pipeline design domains. In the context of Text-to-Image (T2I) workflows specifically, if an adaptation layer or mechanism exists to convert ComfyUI workflows into equivalent workflows for another T2I framework, then the knowledge gained might be transferable due to the shared underlying task.
>
>
> **Q:**
> > "How does the performance of FlowRL scale as the diversity of prompts and flow complexity increases? Is there evidence of reward model generalization in unseen domains or under distribution shifts?"
>
> **A:** Following the reviewer question, we conducted an experiment where we re-trained the reward model using half of our original training set, and evaluated the generalization capabilities of our reward model on the remaining half. We further ensured that some graph structures appear only in the hold-out set, so we can evaluate performance on entirely unseen flow graphs.
>
> The results show good overall performance on the hold-out set, with an $R^2$ score of $0.643$ and a pearson correlation of $0.816$, indicating a strong relationship between predicted and actual values. A more in-depth look shows that: (1) The model generalizes very well to scenarios which contain only parameter or prompt changes compared to what it saw during training (Pearson 0.928). (2) Performance remains good for flows with novel graph structures, but using only seen components (models / blocks) (Pearson 0.667). (3) Performance drops significantly for flows with entirely unseen components, which contain tokens that the reward model has never seen (Pearson 0.152).
>
> After running this experiment, we also evaluated the GRPO stage using the new reward model. The model trained on the full dataset outperforms the model trained on the partial data (71% HPSv2 win-rate), showing the benefit of additional data. Due to time constraints, we were unable to repeat this experiment with additional data amounts, but we would be happy to conduct more such experiments for the revision and present a graph illustrating the scaling of performance as a function of data.

---

### Official Review · Reviewer_mZDq · 2025-07-06

**Clarity:** 2
**Significance:** 2
**Originality:** 2
**Rating:** 4
**Confidence:** 2

**Summary:**

This paper introduces a new framework that uses reinforcement learning to automate the design of complex text-to-image generation pipelines, including a surrogate reward model, two-stage training strategy and dual model guidance. The experimental results demonstrate that the proposed approach not only generates superior visual quality compared to existing methods but also exhibits better diversity.

**Questions:**

- This paper demonstrates that the dual-model guidance technique improves performance by trading off diversity. Have the authors tried dynamically adjusted guidance strength, just like the recent developmenet in CFG[1] to achieve a better balance between quality and diversity ?
- Given the weaknesses discussed above, have the authors considered about how the framework can be extended to add new components ? For instance, could new workflow nodes be added by fine-tuning only a small part of the reward models ?

Reference:

[1]  Analysis of classifier-free guidance weight schedulers

**Ethical Concerns:**

["NO or VERY MINOR ethics concerns only"]

**Final Justification:**

Thank authors for the rebuttal. My concerns have been addressed and I will keep the score.

**Limitations:**

Yes, the authors discuss the limitations in Section 5.

**Quality:**

3

**Strengths And Weaknesses:**

Strengths:
- This paper combines multiple advanced techniques when building the system, including a two-stage training process, first SFT then GRPO, and dual model guidance to further improve generation quality.
- The introduction of surrogate reward model predicts image quality directly from a prompt-workflow pair, which avoids the expensive step of generating an image for every candidate workflow and enables efficient RL training.
- Extensive experimental results are provided, including automated metrics in GenEval and HPSv2, user study, and thorough ablation study.

Weaknesses:
- This paper is actually building an agentic system. However, every time a new model or component is added, a new round of training is needed. This limitation will make the agentic system lose its inherent advantages
- The performance of the entire system, especially the surrogate reward model, is heavily depending on the quality and diversity of the score dataset from ComfyGen. Any biases or gaps in this dataset will accumulate to the final system.

---

> ### Author Rebuttal · Authors · 2025-07-31
>
> We appreciate your review and the points raised regarding our framework's strengths, including the two-stage training, dual model guidance, surrogate reward model, and experimental validation.
>
> We address your weaknesses and questions as follows:
>
> **Q:**
> > "This paper is actually building an agentic system. However, every time a new model or component is added, a new round of training is needed. This limitation will make the agentic system lose its inherent advantages" & "Given the weaknesses discussed above, have the authors considered about how the framework can be extended to add new components ? For instance, could new workflow nodes be added by fine-tuning only a small part of the reward models ?"
>
> **A:** We acknowledge that our current framework will require some retraining as new types of components are introduced and believe this topic to be an excellent direction for future work. We did discuss some pathways for adding more components as they become available, but did not conduct sufficient experiments down this path to offer more than a hypothesis. In general we believe the difficulty of adding new components highly depends on their nature (a new version of an existing model vs. an entirely new type of graph block).
>
> Adding a new version of an existing model (any scenario where the new model is just a parameter change in an existing block, e.g., a new fine-tuned version of SDXL) can likely be done in two stages: (a) It has to be introduced to the surrogate reward model, which can be done by adding a new token (and word embedding) to represent the novel model, and then optimizing the embedding so that the surrogate model correctly predicts the scores of prompt+flow pairs containing the new model. This is similar to existing soft-prompting or personalization techniques. (b) The LLM can be tuned to recognize this component (e.g., with a new embedding + LoRA) by continuing the GRPO training process, with predicted flows occasionally having their models swapped for the new model (and then scored as usual with the surrogate model).
> However, we believe this replacement strategy is unlikely to work for entirely new blocks since dropping them into existing LLM-predicted flows is non-trivial.
>
>
> **Q:**
> > "The performance of the entire system, especially the surrogate reward model, is heavily depending on the quality and diversity of the score dataset from ComfyGen. Any biases or gaps in this dataset will accumulate to the final system."
>
> **A:** We agree that indeed biases or gaps in the dataset could impact the system. However, note that the current dataset already leads to improved performance compared to existing alternatives, and more data could likely lead to additional gains. On the prompt-understanding side, the model uses an LLM backbone which can generalize well to new categories.
>
> Referencing our detailed response to Reviewer 2NbJ03, we also evaluated the reward model's generalization to unseen data by re-training it on a subset of our dataset. We found that while the model can generalize to novel graph structures, performance degrades when they contain entirely unseen blocks or models (e.g., new unseen tokens). However, the model does fine with new graph structures containing the known blocks. Future work could explore active learning or synthetic data generation to further enhance coverage.
>
> **Q:**
> >  "This paper demonstrates that the dual-model guidance technique improves performance by trading off diversity. Have the authors tried dynamically adjusted guidance strength, just like the recent developmenet in CFG[1] to achieve a better balance between quality and diversity"
>
> **A:** In our initial work, we primarily focused on demonstrating the benefits of dual-model guidance using fixed strengths. We did not experiment with dynamically adjusted guidance strength, but are happy to add such experiments to the revision if the reviewer believes them important.

---

### Note · Authors · 2025-08-15

Thank you for the opportunity to respond to the reviews. We are grateful for your time and the insightful feedback provided by the reviewers.

We were particularly pleased to see that reviewers recognized our paper's innovative approach, as "principled step towards end-to-end optimization" (p5tr) and "fresh and significant departure" from prior methods (2NbJ). We were also glad that our introduction of a surrogate reward model was highlighted as a key technical innovation that "effectively eliminate the need for image generation during training" (2NbJ, 9pZC) and provides a "robust optimization strategy" (p5tr).

Furthermore, we were pleased that the paper's empirical validation was recognized as "strong" (2NbJ) and the "robust and thorough ablation analysis" (p5tr, mZDq), was found to provide convincing validation for our method.

We hope that our rebuttal has successfully addressed all of the concerns and clarifications requested. We are confident that our revised paper, informed by this process, is stronger and makes a valuable contribution to the field.

We appreciate your time and consideration.

---

### Decision · Program_Chairs · 2025-09-17

**Decision:**

Accept (poster)

**Comment:**

In this the paper, a reinforcement learning based method FlowRL is proposed to build an agent to automate the workflow within the ComfyUI ecosystem for text-to-image generation. The FlowRL consists of two steps, a SFT step to finetune LLM to predict workflow from a given prompt with supervised pairs, and a RL step with GRPO to finetune LLM to predict workflow with a surrogate reward (more specifically, the ensemble of multiple surrogate rewards to mitigate reward hacking).

Overall, I feel this approach of building an agent to automate the T2I generation workflow via RL with  surrogate reward is quite interesting, and has some novelty.  Even though this submission has some limitations like the proposed approach heavily depends on ComfyGen ecosystem, challenges to add new components/models, lack of methodology novelty (applying existing SFT/GRPO methods to T2I generation workflow agent), however, I still believe this paper is setting a new and important problem of T2I generation workflow agent, and proposed one of the first solutions and valuable insights.

So I would recommend "Accept (poster)".